# Elzaki residual power series method to solve fractional diffusion equation

**Rajendra Pant[1], Geeta Arora[1], Homan Emadifar** [ORCID][2,3,4]*

**1** Department of Mathematics, School of Chemical Engineering and Physical Sciences, Lovely Professional University, Punjab, India, **2** Department of Mathematics, Saveetha School of Engineering, Saveetha Institute of Medical and Technical Sciences, Saveetha University, Chennai, Tamil Nadu, India, **3** MEU Research Unit, Middle East University, Amman, Jordan, **4** Department of Mathematics, Hamedan Branch, Islamic Azad University, Hamedan, Iran

* homan_emadi@yahoo.com

**Data Availability Statement:** The data used to support the findings of this study are included within the article.

**Funding:** The author(s) received no specific funding for this work.

## Abstract

The time-fractional order differential equations are used in many different contexts to analyse the integrated scientific phenomenon. Hence these equations are the point of interest of the researchers. In this work, the diffusion equation for a one-dimensional time-fractional order is solved using a combination of residual power series method with Elzaki transforms. The residual power series approach is a useful technique for finding approximate analytical solutions of fractional differential equations that needs the residual function's $(n-1)\alpha$ derivative. Since it is challenging to determine a function's fractional-order derivative, the traditional residual power series method's application is somewhat constrained. The Elzaki transform with residual power series method is an attempt to get over the limitations of the residual power series method. The obtained numerical solutions are compared with the exact solution of this equation to discuss the method's applicability and efficiency. The results are also graphically displayed to show how the fractional derivative influences the behaviour of the solutions to the suggested method.

## 1. Introduction

Fractional differential equations (FDEs) has broad implications and wide range of applications in various types of problems arising in signal processing systems, diffusion-reaction processes, and electrical network systems [1] etc. These equations have attracted the attention of researchers to study the effect of the fractional derivative involved. The FDEs [2], which have been extensively used over the past few decades in a variety of scientific fields, are extended versions of the classical differential equations. Despite the fact that there are numerous numerical and analytical methods for mathematically solving FDEs, researchers are working to develop new methods that could result in a more accurate solution of the fractional equations.

There are various reliable, popular as well as efficient numerical and analytic methods for finding the solutions of fractional order differential problems. Out of them residual power series method [3], homotopy perturbation and analysis methods [4], differential transform method [5], iterative method [6], Adomian decomposition method [7], different forms of

**Competing interests:** The authors have declared that no competing interests exist

non-integer power series version [8], biological engineering image dispensation [9], physical model [10], risk analysis [11], Taylor's method [12], novel hybrid D(TQ) method [13] and so many other methods have been used for solving fractional order linear or non-linear differential equations [14].

One reliable and efficient method for solving non-integer order differential equations is the residual power series approach [15]. Finding the series solutions and coefficients for non-linear non-integer differential equations is both impossible and challenging. Utilising transformed functions as recurrence relations, solutions can be obtained with a residual power series technique to obtain the coefficients in sequential form. The residual power series approach involves differentiation of the n[th] partial sum of that power series (n-1) times to determine its n[th] ordered coefficients solutions. Ordinary derivatives are generally updated to fractional derivatives for solving non-integer non-linear issues [16].

The fractional order differential equations [17] can be solved using a variety of transforms, such as the Laplace, Sumudu, Elzaki and many more transforms. Elzaki transform is one of the transforms that has been utilised in finding the numerical solution of the well-known differential equations that exist as mathematical models of the phenomena occurring in science and technology [18].

Researchers have used a variety of transforms along with well-known methodologies to solve differential equations, including non-integer order logistic differential models [19], non-integer order BBM-Burger equations [20], non-integer order relaxation-oscillation differential equations [21], and more. The Elzaki transform with residual power series method (ERPSM) has been successfully used to solve some well-known differential equations of non-integer order. The ERPSM was developed to provide for the analytical approximate solution of non-integer order differential models that exists for a variety of applications in mathematics, physics, and engineering. Using the Elzaki transform with residual power series method to solve two-dimensional methodology is the main goal of the current effort, which aims to increase the method's accuracy [22]. The Elzaki transform is a modified version of the Laplace transform and Sumudu transform.

Numerous fractional generalisations of the time-fractional diffusion equation have been put forth over the course of the last few decades and have generated a great deal of controversy in both the academic literature and various diffusion model implementations. The time-fractional diffusion equation is a partial differential equation that incorporates the concepts of fractional calculus to describe the temporal evolution of a variable, such as heat, mass, or particles. The time-fractional diffusion equation adds fractional derivatives [3] in the temporal domain as opposed to the usual diffusion equation, which uses integer-order derivatives. This property makes the model particularly useful for studying phenomena with complex temporal dependencies or long-range interactions because it allows the model to accurately reflect non-local and memory-dependent behaviours in diffusion processes [23]. Although numerous numerical and analytical methods have been used to solve the fractional diffusion equations. The ERPSM is attempted to be used in this case to solve the equation for different values of the fractional power.

The two-dimensional non-integer order diffusion equation is,

$$D_t^\alpha u(x, y, t) = u_{xx}(x, y, t) + u_{yy}(x, y, t) \text{ with } 0 < \alpha \leq 1 \tag{1}$$

$$\text{with initial condition, } u(x, y, 0) = sinx \, siny \tag{2}$$

and exact solution is, $u(x, y, t) = e^{-2t} \, sinxsiny$ for $\alpha = 1$

$$\text{For } 0 < \alpha < 1 \text{ exact solution is, } u(x, y, t) = E_\alpha(z) \, sinxsiny \tag{3}$$

where $E_\alpha(z) = \sum_{k=0}^\infty \frac{z^k}{\Gamma(1+k\alpha)}, z = -4it^\alpha$

The structure has been implemented as follows: An introduction is given in the opening section. The second section provides preliminaries. The research technique of the ERPSM for solving the two-dimensional non-integer order diffusion equation is outlined in the third section. The fourth section provides the numerical experiment to this equation using ERPSM. The fifth section contains graphs and numerical simulations of this equation with various terms. The research topic's conclusion is finally discussed in the sixth part.

## 2. Preliminaries

A powerful tool for characterising the memory of various substances and the nature of inheritance are fractional order differential equations, which are generalised and non-integer order differential equations that can be obtained in time and space with a power law memory kernel of the nonlocal relationships. These investigations all have a well-defined physical foundation, which creates a new avenue for scientific investigation that includes numerical techniques and theoretical analysis for fractional dynamical systems. Fractional differential equations (FDEs), deal with fractional derivatives of the type $\frac{d^\alpha}{dx^\alpha}$ where $0 < \alpha \leq 1$ that are specified for $0 < \alpha$ where $\alpha$ is not always an integer. These represent extensions of the standard differential equations to a noninteger, random order. In last few years the capacity of FDEs to simulate complicated processes has drawn a lot of attention.

### Riemann–Liouville FDE

An FDE of the Riemann–Liouville type is defined as $D^\alpha f(x) = u(x, f(x))$. For this kind of FDE, the initial conditions take the following form:

$D^{\alpha\text{-}k} f(0) = b_k$, k = 1, 2,. . ., n-1
$I^{n-\alpha} f(0) = b_n$.

Similarly, the equation of the form $D_t^\beta u(x, t) + Au(x, t) = (x, f(x))$ with initial condition $D^k u(x,0) = b_k$ is called FDE of Caputo type.

Let us examine the FDE that utilises the Riemann–Liouville derivative.

The fractional derivative $D_t^\alpha$ known as Riemann- Liouville fractional derivative with respect to $t$ can be defined as:

$$D_t^\alpha \mathrm{u}(\mathrm{t}) = \frac{1}{\Gamma(n-\alpha)} \frac{d^n}{dt^n} \int_0^t (t-\tau)^{n-\alpha-1} \mathrm{u}(\tau)\, \mathrm{d}\tau \quad n-1 < \alpha < n$$

$$= u^n(\mathrm{t}) \qquad\qquad\qquad\qquad \alpha = \mathrm{n} \in \mathbb{N},$$

### Mittag-Leffler function

The **Mittag-Leffler function** a unique complex function denoted as $E_{\alpha,\beta}(z)$ depends on two complex parameters α and β is defined as follows: $E_{\alpha,\beta}(z) = \sum_{i=0}^\infty \frac{z^i}{\Gamma(\alpha i + \beta)}$

Where $\Gamma$ is a Gamma function and the real part $\alpha$ is strictly positive.

### Laplace transform for fractional derivative

Laplace transform of fractional derivatives property is,

$$\mathcal{L}\left[D_t^\alpha u(x, t)\right] = s^\alpha \mathcal{L}[u(x, t)] - s^{\alpha-1} u(x, 0)$$

The Laplace transform is not dependent on the beginning value of f, which is usually provided in physical applications, but rather on the initial value of the fractional integral of f. It is commonly recognised that additional conditions must be specified in order to provide a unique solution while solving classical and FDEs. The Caputo derivative of the fractional derivative is used to provide a solution to this problem, and these extra criteria are essentially the standard conditions that are comparable to those of the well-known classical differential equations. Therefore, in most situations, the equation of choice is the one based on the Caputo derivative, which includes the function's initial values as well as its lower-order integer derivatives.

## Elzaki transform

Elzaki transform of function f(t) is denoted by E(f(t)) and is defined as,

$$E(f(t)) = u^2 \int_0^\infty f(ut)e^{-t}dt \qquad u \in (k_1, k_2)$$

The traditional Fourier integral is the source of the Elzaki Transform. The process of solving ordinary, partial and fractional differential equations easier in the time domain easier, Tarig Elzaki created the Elzaki transform. The Fourier, Laplace, and Sumudu transforms are commonly utilised as practical mathematical methods for solving differential equations. Additionally, the Elzaki transform and several of its basic features are employed in this process.

## Some important formulae of Elzaki transform

1. $E(1) = \vartheta^2$

2. $E(t) = \vartheta^3$

3. $E(t^n) = n!\vartheta^{n+2}$

4. $E(e^{at}) = \frac{\vartheta^2}{1-a\vartheta}$

Since the unity preservation characteristics of the Elzaki transform have been demonstrated, problems may be solved using this method. As can be seen in this paper, it is one of the benefits of this new transform, especially in applications to selected equation with its property. In actuality, the linearity of the Elzaki transform is preserved.

## Elzaki transform of fractional derivative

The Elzaki transform of Caputo fractional derivative is defined as,

$$E\left[D_t^\alpha f(t)\right] = \vartheta^{-\alpha}E[f(t)] - \sum_{\beta=0}^{n-1} \vartheta^{2-\alpha+\beta}f^{(\beta)}(0) \text{ where } 1 - \frac{1}{n} < \alpha < 1.$$

## Convergence

Another important concept related to the series solution is its convergence. A series of the form, $\sum_{n=0}^\infty c_n(x - x_0)^n = c_0 + c_1(x - x_0) + c_2(x - x_0)^2 + \cdots$ is known as a general power series in $x-x_0$. In particular, an infinite series $\sum_{n=0}^\infty c_n x^n = c_0 + c_1 x + c_2 x^2 + \cdots$ is known as power series in $x$. The power series $\sum_{n=0}^\infty c_n(x - x_0)^n$ converges (absolutely) for $|x| < R$ where $R = \lim_{n \to \infty} \left|\frac{c_n}{c_{n+1}}\right|$, provided that the limit exists. For convergence of series, it must be infinite

series but the numerical experiment of diffusion equation by ERPSM is a finite series and hence it must be convergence.

The convergence of residual power series method is presented by [24] as the following theorem:

Suppose that $x_i(t)$ are the exact solution for system of initial value problems. Then, the approximate solutions obtained by the residual power series technique are the Taylor expansion of $x_i(t)$. This Taylor expansion of $x_i(t)$ is convergent.

## 3. Methodology

The steps that make up the methodology for solving the fractional diffusion equation using ERPSM [25] are as follows:

**Step 1** Applying Elzaki transform on Eq (1) as,

$$E\big(D_t^\alpha u(x, y, t)\big) = E\Big(u_{xx}(x, y, t) + u_{yy}(x, y, t)\Big) \tag{4}$$

Applying the differentiation property of Elzaki transform,
$E\big[D_t^\alpha u(x, y, t)\big] = \frac{1}{v^\alpha}\{E(u(x, y, t)) - g(x, y, t)\}$ on Eq (4), we get

$$\frac{1}{v^\alpha}\{E(u(x, y, t)) - g(x, y, 0)\} = E\Big\{(u(x, y, t))_{xx} + (u(x, y, t))_{yy}\Big\}$$

$$i.e. E(u(x, y, t)) = g(x, y, 0) + v^\alpha E\Big\{(u(x, y, t))_{xx} + (u(x, y, t))_{yy}\Big\} \tag{5}$$

**Step 2** Taking inverse Elzaki transform in Eq (5), we get

$$u(x, y, t) = G(x, y, 0) + E^{-1}\Big[v^\alpha E\Big\{(u(x, y, t))_{xx} + (u(x, y, t))_{yy}\Big\}\Big] \tag{6}$$

here G(x, y, 0) is the initial condition of the given equation.

**Step 3** By this method algorithm of $u(x,y,t)$ is proposed as,

$$u(x, y, t) = \sum_{n=0}^{\infty} f_n(x, y) \frac{t^{n\alpha}}{(n\alpha)!} \tag{7}$$

To find the mathematical solutions of (7), $u_i(x,y,t)$ may be written in the form,

$$s_i = \sum_{n=0}^{i} u_n(x, y, t) = \sum_{n=0}^{i} f_n(x, y) \frac{t^{n\alpha}}{(n\alpha)!} \tag{8}$$

**Step 4** The Elzaki residual function from (6) can be written as,

$$Res_i(x, y, t) = u_i(x, y, t) - G(x, y, 0) - E^{-1}[v^\alpha E\{u_{i-1}(x, y, t))_{xx} + (u_{i-1}(x, y, t))_{yy}\}] \tag{9}$$

and hence the values of $f_n(x,y)$ may be obtained by putting $n = 0,1,2,\ldots\ldots$ in the relation,

$$t^{-n\alpha} Res_n(x, y, t)/_{t=0} = 0 \tag{10}$$

The pseudo code of the methodology can be demonstrated as in Fig 1.

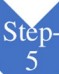

Step-1 • Apply Elzaki transform on fractional diffusion equation defined in equation (1).

Step-2 • Take inverse Elzaki on that diffusion equation (Refer equation (6)).

Step-3 • Frame the algorithm of u(x, y, t) by Elzaki residual power series approach.

Step-4 • Define Elzaki residual function of this method as in equation (9).

Step-5 • The required coefficients $f_n(x, y)$ can thus be obtained by using the relation (10). Substituting in equation (7) for finite term gives the required solution.

**Fig 1. Pseudo code of the methodology.**

## 4. Numerical experiment

Using Elzaki transform on Eq (1), then

$$E\big(D_t^\alpha u(x, y, t)\big) = E\Big(u_{xx}(x, y, t) + u_{yy}(x, y, t)\Big) \tag{11}$$

Applying the differentiation property of Elzaki transforms and so this equation becomes,

$$E(u(x, y, t)) = g(x, y, 0) + v^\alpha E\Big\{(u(x, y, t))_{xx} + (u(x, y, t))_{yy}\Big\} \tag{12}$$

Taking inverse Elzaki transform in Eq (12), we get

$$u(x, y, t) = G(x, y, 0) + E^{-1}\Big[v^\alpha E\Big\{(u(x, y, t))_{xx} + (u(x, y, t))_{yy}\Big\}\Big] \tag{13}$$

Now $u_i(x,y,t)$ may be written in the form,

$$s_i = \sum_{n=0}^{i} u_n(x, y, t) = \sum_{n=0}^{i} f_n(x, y) \frac{t^{n\alpha}}{(n\alpha)!} \tag{14}$$

Then the values of $f_n(x,y)$ can be obtained by using

$$Res_i(x, y, t) = u_i(x, y, t) - G(x, y, 0) - E^{-1}[v^\alpha E\{u_{i-1}(x, y, t))_{xx} + (u_{i-1}(x, y, t))_{yy}\}] \tag{15}$$

When $i = 0$ from Eq (15),
$Res_0(x, y, t) = u_0(x, y, 0) - G(x, y, 0)$ and from Eq (14),

$$0 = u_0(x, y, 0) - G(x, y, 0) \text{i.e.} u_0(x, y, 0) = G(x, y, 0)$$

$$u_0(x, y, 0) = G(x, y, 0) = f_0(x, y) = sinxsiny \tag{16}$$

When $i = 1$ from Eq (15),
$Res_1(x, y, t) = u_1(x, y, t) - G(x, y, 0) - E^{-1}\Big[v^\alpha E\Big\{(u_0(x, y, t))_{xx} + (u_0(x, y, t))_{yy}\Big\}\Big]$ with the conditions $u_1(x, y, t) = f_0(x, y) + f_1(x, y)\frac{t^\alpha}{\alpha!}$ then we can obtain,

$$Res_1(x, y, t) = f_0(x, y) + f_1(x, y)\frac{t^\alpha}{\alpha!} - G(x, y, 0) - E^{-1}\left[v^\alpha E\{f_0(x, y))_{xx} + f_0(x, y))_{yy}\}\right]$$

$$= sinxsiny + f_1(x, y)\frac{t^\alpha}{\alpha!} - sinxsiny - E^{-1}\left[v^\alpha E\left\{(sinxsiny)_{xx} + (sinxsiny)_{yy}\right\}\right]$$

$$= f_1(x, y)\frac{t^\alpha}{\alpha!} - E^{-1}[v^\alpha E(-sinxsiny - sinxsiny)]$$

$$= f_1(x, y)\frac{t^\alpha}{\alpha!} - E^{-1}[v^\alpha E(-2sinxsiny)]$$

$$= f_1(x, y)\frac{t^\alpha}{\alpha!} + 2sinxsiny E^{-1}[v^\alpha E(1)]$$

$$= f_1(x, y)\frac{t^\alpha}{\alpha!} + 2sinxsiny E^{-1}[v^{\alpha+2}]$$

$$= f_1(x, y)\frac{t^\alpha}{\alpha!} + 2sinxsiny\frac{t^\alpha}{\alpha!}$$

$$= \{f_1(x, y) + 2sinxsiny\}\frac{t^\alpha}{\alpha!}$$

Then after solving $t^{-\alpha}Res_1(x, y, t)_{t=0} = 0$ gives that

$$f_1(x, y) + 2\ sinxsiny = 0\ i.\ e.f_1(x, y) = -2sinxsiny \tag{17}$$

When $i = 2$ from Eq (15)

$Res_2(x, y, t) = u_2(x, y, t) - G(x, y, 0) - E^{-1}\left[v^\alpha E\left\{(u_1(x, y, t))_{xx} + (u_1(x, y, t))_{yy}\right\}\right]$, with conditions $u_1(x, y, t) = f_0(x, y) + f_1(x, y)\frac{t^\alpha}{\alpha!}$ and

$$u_2(x, y, t) = f_0(x, y) + f_1(x, y)\frac{t^\alpha}{\alpha!} + f_2(x, y)\frac{t^{2\alpha}}{(2\alpha)!}, \text{ we get}$$

$$Res_2(x, y, t) = f_0(x, y) + f_1(x, y)\frac{t^\alpha}{\alpha!} + f_2(x, y)\frac{t^{2\alpha}}{(2\alpha)!} - f_0(x, y)$$

$$- E^{-1}\left[v^\alpha E\left\{\left(f_0(x, y) + f_1(x, y)\frac{t^\alpha}{\alpha!}\right)_{xx} + \left(f_0(x, y) + f_1(x, y)\frac{t^\alpha}{\alpha!}\right)_{yy}\right\}\right]$$

$$= f_1(x, y)\frac{t^\alpha}{\alpha!} + f_2(x, y)\frac{t^{2\alpha}}{(2\alpha)!}$$

$$- E^{-1}\left[v^\alpha E\left\{\left(sinxsiny - 2sinxsiny\frac{t^\alpha}{\alpha!}\right)_{xx} + (sinxsiny - 2sinxsiny\frac{t^\alpha}{\alpha!})_{yy}\right\}\right]$$

$$= -2sinxsiny\frac{t^\alpha}{\alpha!} + f_2(x, y)\frac{t^{2\alpha}}{(2\alpha)!}$$

$$- E^{-1}\left[v^\alpha E\left\{-sinxsiny + 2sinxsiny\frac{t^\alpha}{\alpha!} + -sinxsiny + 2sinxsiny\frac{t^\alpha}{\alpha!}\right\}\right]$$

$$= -2sinxsiny\frac{t^\alpha}{\alpha!} + f_2(x, y)\frac{t^{2\alpha}}{(2\alpha)!} - E^{-1}\left[v^\alpha E\left\{-2sinxsiny + 4sinxsiny\frac{t^\alpha}{\alpha!}\right\}\right]$$

$$= -2sinxsiny\frac{t^\alpha}{\alpha!} + f_2(x, y)\frac{t^{2\alpha}}{(2\alpha)!} - E^{-1}\left[v^\alpha E\left\{-2sinxsiny + 4sinxsiny\frac{t^\alpha}{\alpha!}\right\}\right]$$

$$= -2sinxsiny\frac{t^\alpha}{\alpha!} + f_2(x, y)\frac{t^{2\alpha}}{(2\alpha)!} - E^{-1}\left[v^\alpha\{-2sinxsiny\, E(1) + 4sinxsinyE\left(\frac{t^\alpha}{\alpha!}\right)\right]$$

$$= -2sinxsiny\frac{t^\alpha}{\alpha!} + f_2(x, y)\frac{t^{2\alpha}}{(2\alpha)!} - E^{-1}\left[v^\alpha\{-2sinxsiny\, v^2 + 4\, sinxsiny\frac{\alpha!v^{\alpha+2}}{\alpha!}\right]$$

$$= -2sinxsiny\frac{t^\alpha}{\alpha!} + f_2(x, y)\frac{t^{2\alpha}}{(2\alpha)!} + 2\, sinxsinyE^{-1}(v^{\alpha+2}) - 4\, sinxsiny\, E^{-1}[v^{2\alpha+2}]$$

$$= -2\, sinxsiny\frac{t^\alpha}{\alpha!} + f_2(x, y)\frac{t^{2\alpha}}{(2\alpha)!} + 2\, sinxsiny\frac{t^\alpha}{\alpha!} - 4\, sinxsiny\frac{t^{2\alpha}}{(2\alpha)!}$$

$$= \{f_2(x, y) - 4\, sinxsiny\}\frac{t^{2\alpha}}{(2\alpha)!}$$

Therefore, from $t^{-2\alpha}Res_2(x, y, t)/_{t=0} = 0$ we have,

$$f_2(x, y) - 4\, sinxsiny = 0\ i.\ e.f_2(x, y) = 4sinxsiny \tag{18}$$

Now, the second approximate solution is,

$$u_2(x, y, t) = sinxsiny - 2sinxsiny\frac{t^\alpha}{\alpha!} + 4sinxsiny\frac{t^{2\alpha}}{(2\alpha)!}$$

When $i = 3$ from Eq (15)

$$Res_3(x, y, t) = u_3(x, y, t) - G(x, y, 0) - E^{-1}\left[v^\alpha E\Big\{(u_2(x, y, t))_{xx} + (u_2(x, y, t))_{yy}\Big\}\right], \text{ with}$$

conditions $u_2(x, y, t) = f_0(x, y) + f_1(x, y)\frac{t^\alpha}{\alpha!} + f_2(x, y)\frac{t^{2\alpha}}{(2\alpha)!}$ and

$u_3(x, y, t) = f_0(x, y) + f_1(x, y)\frac{t^\alpha}{\alpha!} + f_2(x, y)\frac{t^{2\alpha}}{(2\alpha)!} + f_3(x, y)\frac{t^{3\alpha}}{(3\alpha)!}$ we get,

$$Res_3(x, y, t) = f_0(x, y) + f_1(x, y)\frac{t^\alpha}{\alpha!} + f_2(x, y)\frac{t^{2\alpha}}{(2\alpha)!} + f_3(x, y)\frac{t^{3\alpha}}{(3\alpha)!} - f_0(x, y)$$

$$- E^{-1}\left[v^\alpha E\left\{\left(f_0(x, y) + f_1(x, y)\frac{t^\alpha}{\alpha!} + f_2(x, y)\frac{t^{2\alpha}}{(2\alpha)!}\right)_{xx} + \left(f_0(x, y) + f_1(x, y)\frac{t^\alpha}{\alpha!} + f_2(x, y)\frac{t^{2\alpha}}{(2\alpha)!}\right)_{yy}\right\}\right]$$

$$= -2sinxsiny\frac{t^\alpha}{\alpha!} + 4sinxsiny\frac{t^{2\alpha}}{(2\alpha)!} + f_3(x, y)\frac{t^{3\alpha}}{(3\alpha)!}$$

$$- E^{-1}\left[v^\alpha E\{(sinxsiny - 2sinxsiny\frac{t^\alpha}{\alpha!} + 4sinxsiny\frac{t^{2\alpha}}{(2\alpha)!})_{xx} + \left(sinxsiny - 2sinxsiny\frac{t^\alpha}{\alpha!} + 4sinxsiny\frac{t^{2\alpha}}{(2\alpha)!}\right)_{yy}\}\right]$$

$$= -2sinxsiny\frac{t^\alpha}{\alpha!} + 4ssinxsiny\frac{t^{2\alpha}}{(2\alpha)!} + f_3(x, y)\frac{t^{3\alpha}}{(3\alpha)!}$$

$$- E^{-1}\left[v^\alpha E\{-sinxsiny + 2sinxsiny\frac{t^\alpha}{\alpha!} - 4sinxsiny\frac{t^{2\alpha}}{(2\alpha)!} - sinxsiny + 2sinxsiny\frac{t^\alpha}{\alpha!} - 4sinxsiny\frac{t^{2\alpha}}{(2\alpha)!}\}\right]$$

$$= -2\,sinxsiny\frac{t^\alpha}{\alpha!} + 4sinxsiny\frac{t^{2\alpha}}{(2\alpha)!} + f_3(x, y)\frac{t^{3\alpha}}{(3\alpha)!}$$

$$- E^{-1}\left[v^\alpha E\{-2sinxsiny + 4sinxsiny\frac{t^\alpha}{\alpha!} - 8sinxsiny\frac{t^{2\alpha}}{(2\alpha)!}\}\right]$$

$$= -2sinxsiny\frac{t^\alpha}{\alpha!} + 4sinxsiny\frac{t^{2\alpha}}{(2\alpha)!} + f_3(x, y)\frac{t^{3\alpha}}{(3\alpha)!}$$

$$- E^{-1}\left[v^\alpha\{-2sinxsiny\,E(1) + 4sinxsiny\frac{E(t^\alpha)}{\alpha!} - 8sinxsiny\frac{E(t^{2\alpha})}{(2\alpha)!}\}\right]$$

$$= -2sinxsiny\frac{t^\alpha}{\alpha!} + 4sinxsiny\frac{t^{2\alpha}}{(2\alpha)!} + f_3(x, y)\frac{t^{3\alpha}}{(3\alpha)!}$$

$$- E^{-1}\left[v^\alpha\left\{-2sinxsiny\,v^2 + 4sinxsiny\frac{\alpha!v^{\alpha+2}}{\alpha!} - 8sinxsiny\frac{(2\alpha)!v^{2\alpha+2}}{(2\alpha)!}\right\}\right]$$

$$= -2sinxsiny\frac{t^\alpha}{\alpha!} + 4sinxsiny\frac{t^{2\alpha}}{(2\alpha)!} + f_3(x,y)\frac{t^{3\alpha}}{(3\alpha)!}$$
$$- E^{-1}[-2sinxsiny^{\alpha+2} + 4sinxsiny^{2\alpha+2} - 8sinxsiny\,v^{3\alpha+2}]$$

$$= -2sinxsiny\frac{t^\alpha}{\alpha!} + 4sinxsiny\frac{t^{2\alpha}}{(2\alpha)!} + f_3(x,y)\frac{t^{3\alpha}}{(3\alpha)!} + 2sinxsiny\,E^{-1}(v^{\alpha+2})$$
$$- 4sinxsiny\,E^{-1}(v^{2\alpha+2}) + 8sinxsiny\,E^{-1}(v^{3\alpha+2})$$

$$= -2sinxsiny\frac{t^\alpha}{\alpha!} + 4sinxsiny\frac{t^{2\alpha}}{(2\alpha)!} + f_3(x,y)\frac{t^{3\alpha}}{(3\alpha)!} + 2sinxsiny\frac{t^\alpha}{\alpha!} - 4sinxsiny\frac{t^{2\alpha}}{(2\alpha)!}$$
$$+ 8sinxsiny\frac{t^{3\alpha}}{(3\alpha)!}$$

$$= f_3(x,y)\frac{t^{3\alpha}}{(3\alpha)!} + 8sinxsiny\frac{t^{3\alpha}}{(3\alpha)!}$$

$$= \{f_3(x,y) + 8sinxsiny\}\frac{t^{3\alpha}}{(3\alpha)!}$$

Therefore, from $t^{-3\alpha}Res_3(x,y,t)/_{t=0} = 0$ we have,

$$f_3(x,y) + 8sinxsiny = 0 \; i.e. f_3(x,y) = -8sinxsiny \tag{19}$$

Now, the third approximate solution is,

$$u_3(x,y,t) = sinxsiny - 2sinxsiny\frac{t^\alpha}{\alpha!} + 4sinxsiny\frac{t^{2\alpha}}{(2\alpha)!} - 8sinxsiny\frac{t^{3\alpha}}{(3\alpha)!}$$

Similarly, the $n^{th}$ coefficient of $u(x,y,t)$ is $f_n(x,y) = (-2)^n sinxsiny$ At last the $n^{th}$ ERPSM approximate solutions of $u(x,y,t)$ is

$$u_n(x,y,t) = sinxsiny\sum_{n=0}^{i}\frac{(-2t^\alpha)^n}{(n\alpha)!} \tag{20}$$

## 5. Numerical simulations and graphs

The numerical solution of this equation has been obtained for the domain [0, 1] for both x and y and the results are presented at t = 1 for α = 0.5, 0.8 and 1.0 in **Tables 1–3** and **Figs 2–4.** **Tables 4 and 5** shows that a comparison of the maximum errors of diffusion equation by ERPSM and by RPSM [26] for α = 1 and with the present approach taking different number of terms. This verified that the considered method is suitable and reliable for the solution for the fractional differential equations. The solution converges rapidly to exact solution when numbers of terms are increased.

## 6. Conclusion

The current research presents the implementation of a novel and trustworthy method ERPSM to numerically solve the non-integer order diffusion equation in two dimensions. This strategy combines the residual power series method, an improvement of the conventional residual power series method, with the Elzaki transform. This method has a benefit of requires less

**Table 1. Solution at t = 1 when value of α = 0.5.**

| y/x | 0.2 | 0.4 | 0.6 | 0.8 | 1.0 |
|---|---|---|---|---|---|
| **0.1** | 0.006805 | 0.013340 | 0.019342 | 0.024573 | 0.028825 |
| **0.2** | 0.013543 | 0.026546 | 0.038491 | 0.048901 | 0.057361 |
| **0.3** | 0.020145 | 0.039487 | 0.057255 | 0.072740 | 0.085325 |
| **0.4** | 0.026546 | 0.052033 | 0.075447 | 0.095852 | 0.112436 |
| **0.5** | 0.032681 | 0.06406 | 0.092885 | 0.118006 | 0.138423 |
| **0.6** | 0.038491 | 0.075447 | 0.109395 | 0.138982 | 0.163028 |
| **0.7** | 0.043915 | 0.086079 | 0.124812 | 0.158569 | 0.186004 |
| **0.8** | 0.048901 | 0.095852 | 0.138982 | 0.176571 | 0.207121 |
| **0.9** | 0.053398 | 0.104667 | 0.151763 | 0.192809 | 0.226168 |
| **1.0** | 0.057361 | 0.112436 | 0.163028 | 0.207121 | 0.242956 |

**Table 2. Solution at t = 1 when value of α = 0.8.**

| y/x | 0.2 | 0.4 | 0.6 | 0.8 | 1.0 |
|---|---|---|---|---|---|
| **0.1** | 0.003764 | 0.007379 | 0.010699 | 0.013593 | 0.015944 |
| **0.2** | 0.007491 | 0.014684 | 0.021291 | 0.027049 | 0.031729 |
| **0.3** | 0.011143 | 0.021842 | 0.03167 | 0.040236 | 0.047197 |
| **0.4** | 0.014684 | 0.028782 | 0.041733 | 0.053020 | 0.062193 |
| **0.5** | 0.018078 | 0.035434 | 0.051379 | 0.065275 | 0.076568 |
| **0.6** | 0.021291 | 0.041733 | 0.060511 | 0.076877 | 0.090178 |
| **0.7** | 0.024291 | 0.047614 | 0.069039 | 0.087711 | 0.102887 |
| **0.8** | 0.027049 | 0.053020 | 0.076877 | 0.097669 | 0.114568 |
| **0.9** | 0.029537 | 0.057896 | 0.083947 | 0.106651 | 0.125104 |
| **1.0** | 0.031729 | 0.062193 | 0.090178 | 0.114568 | 0.134390 |

**Table 3. Solution at t = 1 when value of α = 1.0.**

| y/x | 0.2 | 0.4 | 0.6 | 0.8 | 1.0 |
|---|---|---|---|---|---|
| **0.1** | 0.002684 | 0.005261 | 0.007629 | 0.009692 | 0.011369 |
| **0.2** | 0.005342 | 0.010470 | 0.015182 | 0.019288 | 0.022625 |
| **0.3** | 0.007946 | 0.015575 | 0.022582 | 0.028690 | 0.033654 |
| **0.4** | 0.010470 | 0.020523 | 0.029758 | 0.037806 | 0.044347 |
| **0.5** | 0.012890 | 0.025267 | 0.036636 | 0.046544 | 0.054597 |
| **0.6** | 0.015182 | 0.029758 | 0.043148 | 0.054818 | 0.064302 |
| **0.7** | 0.017321 | 0.033952 | 0.049229 | 0.062543 | 0.073364 |
| **0.8** | 0.019288 | 0.037806 | 0.054818 | 0.069644 | 0.081693 |
| **0.9** | 0.021061 | 0.041283 | 0.059859 | 0.076048 | 0.089206 |
| **1.0** | 0.022625 | 0.044347 | 0.064302 | 0.081693 | 0.095827 |

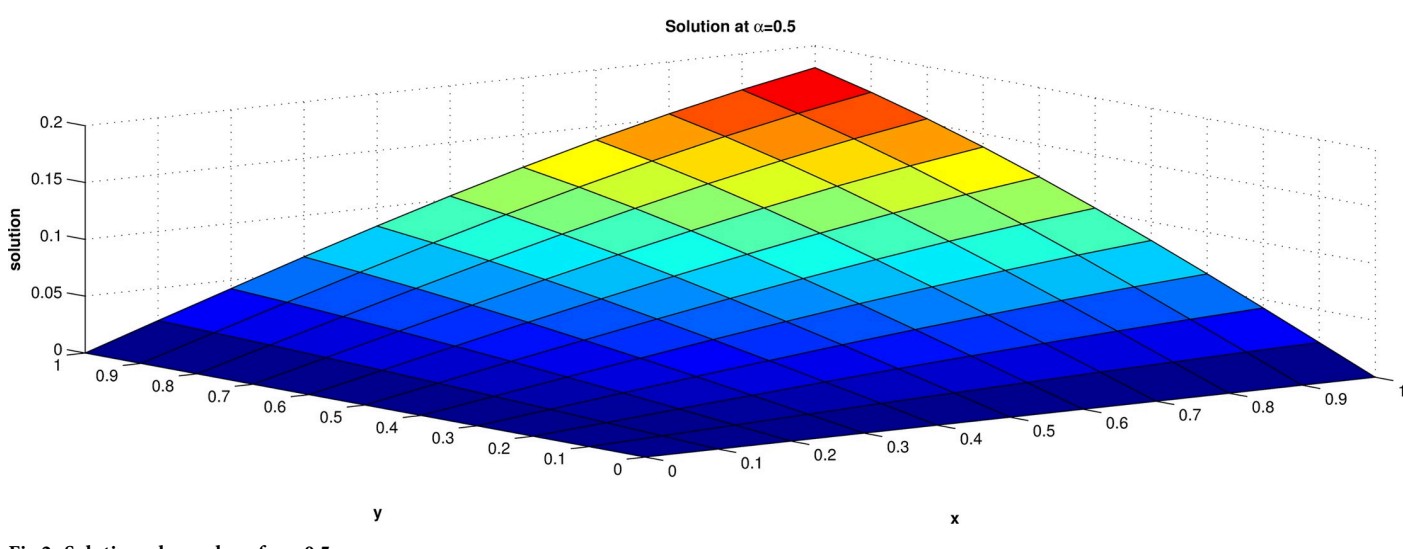

**Fig 2. Solution when value of α = 0.5.**

calculation and give less error in the solution. The aforementioned sequential procedures lead to the determination of the coefficients of this power series solution. ERPSM demonstrated its capacity to solve non-integer order differential equations with sufficient correctness and dependable computing steps for two-dimensional non-integer order diffusion equations. From the table of comparison in numerical example it can be concluded that this novel approach is advantageous. This approach also offers straightforward and precise algorithms for estimating solutions of the fractional order diffusion equations.

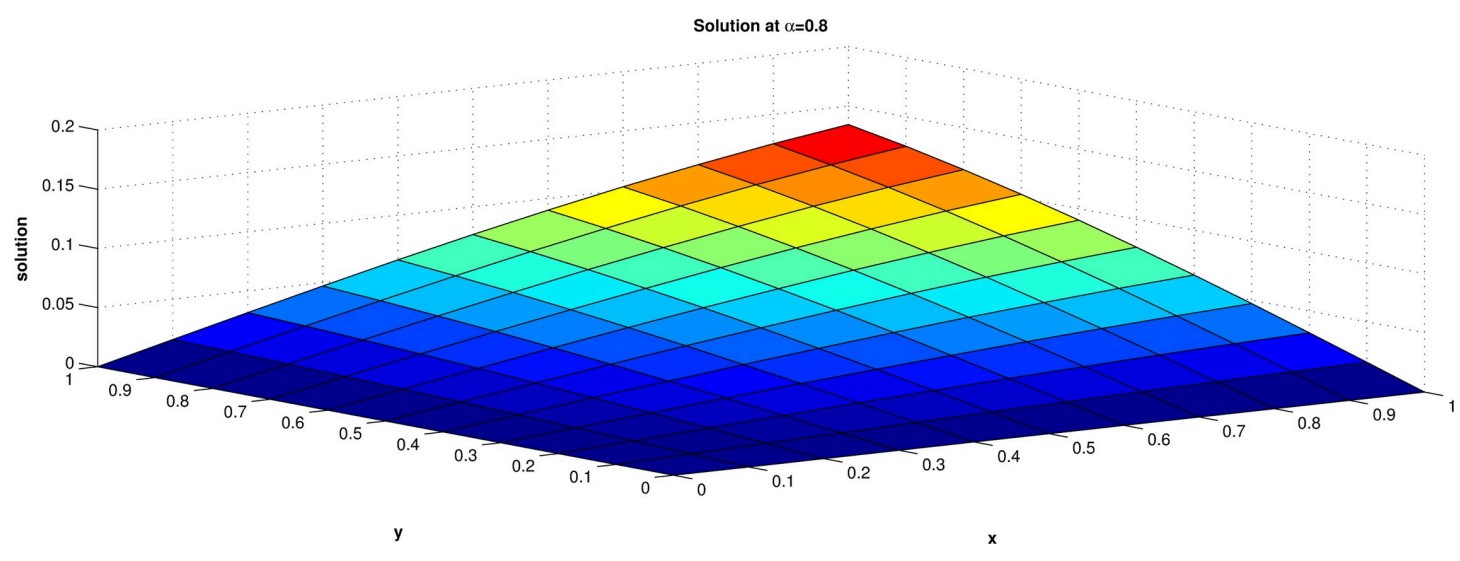

**Fig 3. Solution when value of α = 0.8.**

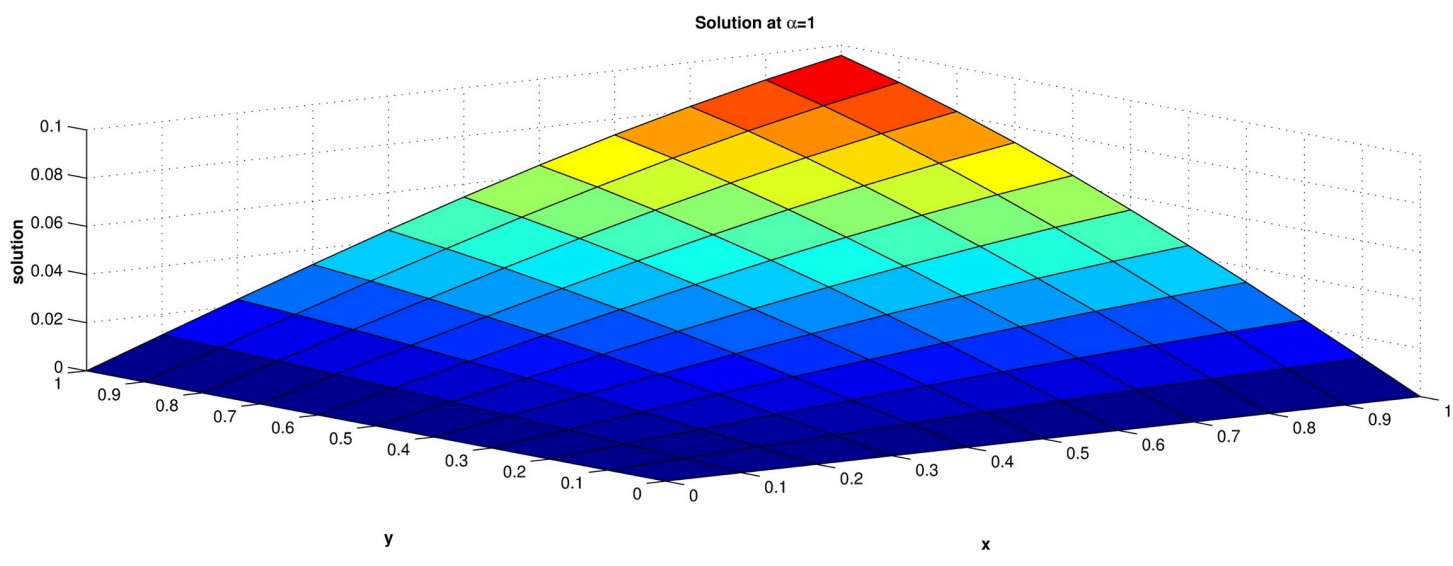

**Fig 4. Solution when value of α = 1.0.**

**Table 4. For α = 1 the maximum error has been presented at different time levels.**

| T | $l_\infty$ |
|---|---|
| 0.1 | 4.4408e-16 |
| 0.2 | 7.1981e-13 |
| 0.3 | 6.1280e-11 |
| 0.4 | 1.4280e-09 |
| 0.5 | 1.6366e-08 |
| 0.6 | 1.1974e-07 |
| 0.7 | 6.4273e-07 |
| 0.8 | 2.7504e-06 |
| 0.9 | 9.8999e-06 |
| 1.0 | 3.1088e-05 |

**Table 5. The comparison of the maximum errors of diffusion equation by ERPSM and RPSM [26] for α = 1.**

| t | ERPSM with number of terms | | | RPSM [26] |
|---|---|---|---|---|
|  | 10 | 12 | 14 |  |
| 0.2 | 7.1981e-13 | 6.1062e-16 | 0 | 0.5e-07 |
| 0.4 | 1.4280e-09 | 5.9121e-12 | 1.8152e-14 | 1.0e-07 |
| 0.6 | 1.1974e-07 | 1.1200e-09 | 7.7582e-12 | 1.5.e-07 |
| 0.8 | 2.7504e-06 | 4.5924e-08 | 5.6724e-10 | 2.5e-06 |
| 1.0 | 3.1088e-05 | 8.1419e-07 | 1.5759e-08 | 3.0e-05 |

## Author Contributions

**Formal analysis:** Geeta Arora.

**Methodology:** Geeta Arora.

**Software:** Geeta Arora.

**Writing – original draft:** Rajendra Pant.

**Writing – review & editing:** Homan Emadifar.

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
