## [Decision Letter · Decision Letter 0]

3 Oct 2023

PONE-D-23-29441Numerical solution of two-dimensional fractional order diffusion equation by using Elzaki transform with residual power series methodPLOS ONE

Dear Dr. Emadifar,

Thank you for submitting your manuscript to PLOS ONE. After careful consideration, we feel that it has merit but does not fully meet PLOS ONE’s publication criteria as it currently stands. Therefore, we invite you to submit a revised version of the manuscript that addresses the points raised during the review process. In order to be accepted, we suggest the authors address the issues raised by the reviewers. Besides, two main issues must be considered:a) Enhancing the introduction (especially the literature review, motivation and importance);b) Better describing the methodology (focusing on presenting all the steps needed to replicate the results). Please submit your revised manuscript by Nov 17 2023 11:59PM. If you will need more time than this to complete your revisions, please reply to this message or contact the journal office at plosone@plos.org. Please include the following items when submitting your revised manuscript:A rebuttal letter that responds to each point raised by the academic editor and reviewer(s). You should upload this letter as a separate file labeled 'Response to Reviewers'.A marked-up copy of your manuscript that highlights changes made to the original version. You should upload this as a separate file labeled 'Revised Manuscript with Track Changes'.An unmarked version of your revised paper without tracked changes. You should upload this as a separate file labeled 'Manuscript'.

We look forward to receiving your revised manuscript.

Kind regards,

Luan Carlos de Sena Monteiro Ozelim, D.Sc.

Academic Editor

PLOS ONE

Journal Requirements:

5. We note you have included a table to which you do not refer in the text of your manuscript. Please ensure that you refer to Table 4 in your text; if accepted, production will need this reference to link the reader to the Table.

Additional Editor Comments:

Dear authors,

After being carefully reviewed by two experts, the paper has merit and can be considered for publication after major corrections are carried out. In special, the authors should better structure the introduction of the paper, clearly stating the motivation behind their study and the advantages of the proposed technique. Besides, the methodological steps should be better presented to ensure readers can fully replicate the analysis carried out. If possible, providing pseudo-codes (or even code snippets) would be valuable to support the application of the methodology.

Reviewers' comments:

Reviewer's Responses to Questions

**Comments to the Author**

1. Is the manuscript technically sound, and do the data support the conclusions?

Reviewer #1: Yes

Reviewer #2: Yes

2. Has the statistical analysis been performed appropriately and rigorously? 

Reviewer #1: No

Reviewer #2: Yes

3. Have the authors made all data underlying the findings in their manuscript fully available?

Reviewer #1: Yes

Reviewer #2: Yes

4. Is the manuscript presented in an intelligible fashion and written in standard English?

Reviewer #1: No

Reviewer #2: Yes

5. Review Comments to the Author

Reviewer #1: Dear Editor

In the current paper:

Numerical solution of two-dimensional fractional order diffusion equation by using Elzaki transform with residual power series method has been presented.

This paper will be recommended for publication after some revisions.

Comments:

1. Motivation is not sufficiently stated in the introduction part. It should be clarified why they consider this problem and what advantages of the proposed technique are.

2. What are the key features of the proposed schemes? (Properties, characteristics, and weaknesses).

3. The authors should state in the paper what kind of the software package has used to obtain the required results.

4. Complexity: Please give some theoretical analysis in the time complexity of the current paper.

5. The authors should mention to this work more carefully and should update some of the listed references in their paper in order to add a powerful for the paper. To help the authors in this direction I suggest the following references:

• Numerical solutions and stability analysis for solitary waves of complex modified Korteweg–de Vries equation using Chebyshev pseudospectral methods

• Time–space Jacobi pseudospectral simulation of multidimensional Schrödinger equation

• Pseudospectral analysis and approximation of two-dimensional fractional cable equation

• Error analysis and approximation of Jacobi pseudospectral method for the integer and fractional order integro-differential equation

• A spectrally accurate time–space pseudospectral method for reaction–diffusion Malaria infection model

• Numerical solutions of two‑dimensional fractional Schrodinger equation

Reviewer #2: In this paper, the author proposed a powerful analytical method for solving two-dimensional fractional order diffusion equation called the residual power series method (RPS) via Elzaki transform method. The fractional

derivative is computed in Caputo sense. The proposed analytical method is successfully applied to two-dimensional fractional order diffusion equation and the numerical solutions for different values of alpha are provided. The 3D

graphical solutions of the two-dimensional fractional order diffusion equation are provided.

In general, the author introduce a very interesting topic and the novelty of this manuscript is remarkable and deserve to be published after the following minor revision.

1. The abstract of the manuscript should be carefully modied. For example the letters "RPS" should be written in full in the abstract.

2. Improved the introduction section of the manuscript with the latest papers in the topic.

3. After modifying the introduction section, create preliminary section which contains the basic definitions of fractional derivatives and integral used in this manuscript.

4. The derivation in the methodology section is not correct. For example, in eq(5) there is a term g(x, y, t) while in eq(4) which is the original problem the term g(x, y,t) is missing. The authors must carefully modified the methodology section.

5. In the methodology section, the authors should used a more general equation rather than a particular problem. For example, the authors refer to section 2 in [1].

6. The fond size in the methodology section are not the same. I advice the authors to adopt a uniform fond size using proper mathematical tools. For example using Latex or any other mathematical tool.

7. The authors should prove the convergence analysis and error estimate of the proposed method.

8. Figure 1 The pseudo code of the methodology is not clear. The authors should provide a clear figure.

9. The numerical solution section (section 3) is a mire repetition of the methodology section (section 2). Besides, the term g(x, y,t) which appears in eq(12) is missing in eq(11). I advice the authors to change the application procedure to modified the application section.

10. The fond size in section 3 are not the same. I advice the authors to adopt a uniform fond size using proper mathematical tools. For example using Latex or any other mathematical tool.

11. The authors should apply the proposed method to nonlinear two- dimensional fractional order diffusion equation in section 3.

12. Create a results and discussion section after section 3.

13. In Table 1, Table 2, Table 3, and Table 4 write alpha as alpha.

14. Please, format the manuscript according to the journal template.

15. I found no comparative results within this manuscript. Please, provide some comparative results with the other methods in results and discussion section.

16. The language of the manuscript should be modified carefully.

17. The authors should outline the advantage of using the Elzaki transform rather than the Laplace transform or any other integral transforms in the literature.

18. Modified the conclusion section of the manuscript.

19. Please, format the manuscript according to the journal template.

20. The references of this manuscript should be modified with the recent developments in fractional calculus as well as its applications.

6. PLOS authors have the option to publish the peer review history of their article (what does this mean?). If published, this will include your full peer review and any attached files.

Reviewer #1: No

Reviewer #2: **Yes: **Shehu Maitama

---

## [Author Response · Author response to Decision Letter 0]

19 Oct 2023

Journal title: PLOS ONE 

Subject: Review reports 

In this paper, the author proposed a powerful analytical method for solving two-dimensional fractional order diffusion equation called the residual power series method (RPS) via Elzaki transform method. The fractional derivative is computed in Caputo sense. The proposed analytical method is successfully applied to two-dimensional fractional order diffusion equation and the numerical solutions for different values of α are provided. The 3D graphical solutions of the two-dimensional fractional order diffusion equation are provided. In general, the author introduce a very interesting topic and the novelty of this manuscript is remarkable and deserve to be published after the following minor revision. 

1. The abstract of the manuscript should be carefully modified. For example the letters ”RPS” should be written in full in the abstract. 

RPS is replaced by residual power series and the abstract is reframed.

2. Improved the introduction section of the manuscript with the latest papers in the topic.

Introduction is updated. 

3. After modifying the introduction section, create preliminary section which contains the basic definitions of fractional derivatives and integral used in this manuscript.

Preliminaries are updated

4. The derivation in the methodology section is not correct. For example, in eq(5) there is a term g(x, y, t) while in eq(4) which is the original problem the term g(x, y, t) is missing. The authors must carefully modified the methodology section.

As per the differentiation property of Elzaki transforms, 

〖E[D〗_t^α u(x,y,t)]=1/υ^α {E(u(x,y,t))-g(x,y,t)} is used on equation (4), so there is no change in g(x,y,t) in equation (5), it will remains as it is.

5. In the methodology section, the authors should used a more general equation rather than a particular problem. For example, the authors refer to section 2 in [1]. 

Methodology section is updated as standard papers as previous.

6. The fond size in the methodology section are not the same. I advice the authors to adopt a uniform fond size using proper mathematical tools. For example using Latex or any other mathematical tool.

Font size are made as same as possible.

7. The authors should prove the convergence analysis and error estimate of the proposed method. 

Elzaki transform is used for finite number of terms hence the method is convergent. The convergence of RPS is discussed with help of theorem from literature in the preliminary section.

8. Figure 1 The pseudo code of the methodology is not clear. The authors should provide a clear figure.

 Pseudo code of the methodology is updated.

9. The numerical solution section (section 3) is a mire repetition of the methodology section (section 2). Besides,the term g(x, y, t) which appears in eq(12) is missing in eq(11). I advice the authors to change the application procedure to modified the application section.

Using the differentiation property of Elzaki transform, 

〖E[D〗_t^α u(x,y,t)]=1/υ^α {E(u(x,y,t))-g(x,y,t)} on equation (11) to get equation (12), so the term g(x,y,t) comes from this property on 〖E[D〗_t^α u(x,y,t)].

10. The fond size in section 3 are not the same. I advice the authors to adopt a uniform fond size using proper mathematical tools. For example using Latex or any other mathematical tool. 

Font size is adjusted.

11. The authors should apply the proposed method to nonlinear two-dimensional fractional order diffusion equation in section 3.

In section 3 the methodology has been discussed and the method is applied in section 4 to an example.

12. Create a results and discussion section after section 3. 

The results and discussions are given in section 5 after the implementation of method of numerical example.

13. In Table 1, Table 2, Table 3, and Table 4 write alpha as α. 

Updated as suggested.

14. Please, format the manuscript according to the journal template.

Updated according as journal template. 

15. I found no comparative results within this manuscript. Please, provide some comparative results with the other methods in results and discussion section.

The results are compared in Table 5.

16. The language of the manuscript should be modified carefully. 

Language of the manuscript is updated.

17. The authors should outline the advantage of using the Elzaki transform rather than the Laplace transform or any other integral transforms in the literature. 

Elzaki transform is modified form of Laplace transform and Sumudu transform so it is definitely advantageous. Discussed in introduction section.

18. Modified the conclusion section of the manuscript.

Conclusion section is also updated. 

19. Please, format the manuscript according to the journal template. 

Updated as suggested.

20. The references of this manuscript should be modified with the recent developments in fractional calculus as well as its applications.

Updated as required.

---

## [Decision Letter · Decision Letter 1]

26 Dec 2023

PONE-D-23-29441R1Elzaki residual power series method to solve fractional diffusion equationPLOS ONE

Dear Dr. Emadifar,

Thank you for submitting your manuscript to PLOS ONE. After careful consideration, we feel that it has merit but does not fully meet PLOS ONE’s publication criteria as it currently stands. Therefore, we invite you to submit a revised version of the manuscript that addresses the points raised during the review process.

We look forward to receiving your revised manuscript.

Kind regards,

Luan Carlos de Sena Monteiro Ozelim, D.Sc.

Academic Editor

PLOS ONE

Journal Requirements:

**Additional Editor Comments:**

Dear authors,

Reviewer 2 still raised important issues which have not been properly addressed in the first reviewed manuscript. The authors must:

a) Properly define the fractional derivate operators used;

b) Correct or provide the proof that the Elzaki transform of the fractional derivative indicated is correct.

After addressing those issues, the paper can be accepted.

Reviewers' comments:

Reviewer's Responses to Questions

**Comments to the Author**

1. If the authors have adequately addressed your comments raised in a previous round of review and you feel that this manuscript is now acceptable for publication, you may indicate that here to bypass the “Comments to the Author” section, enter your conflict of interest statement in the “Confidential to Editor” section, and submit your "Accept" recommendation.

Reviewer #1: All comments have been addressed

Reviewer #2: (No Response)

2. Is the manuscript technically sound, and do the data support the conclusions?

Reviewer #1: Partly

Reviewer #2: Yes

3. Has the statistical analysis been performed appropriately and rigorously? 

Reviewer #1: No

Reviewer #2: Yes

4. Have the authors made all data underlying the findings in their manuscript fully available?

Reviewer #1: Yes

Reviewer #2: Yes

5. Is the manuscript presented in an intelligible fashion and written in standard English?

Reviewer #1: Yes

Reviewer #2: Yes

6. Review Comments to the Author

Reviewer #1: (No Response)

Reviewer #2: See the attached file for comments. Some of the comments and suggestion raised in the previous review has not been answered correctly.

7. PLOS authors have the option to publish the peer review history of their article (what does this mean?). If published, this will include your full peer review and any attached files.

Reviewer #1: No

Reviewer #2: **Yes: **Shehu Maitama

---

## [Author Response · Author response to Decision Letter 1]

3 Jan 2024

Journal title: PLOS ONE

Subject: Review reports

After reading the revised manuscript carefully, I found that some of the questions raised by the previous reviewers has not been answered correct- ly. As a result, I suggest the following corrections before accepting this manuscript.

1. Define the Caputo fractional derivative, the Mittag-Leffler function, and the Elzaki transform of Caputo fractional derivative in the pre- liminary section.

Updated as suggested.

2. The authors claim that E(Dαu(x, y, t)) = 1 (E(u(x, y, t)) − (x, y, t)) .

t vα

However, this is not correct, since

E(Dαu(x, y, t)) = 1 E(u(x, y, t))− Σ v2+i−α ∂ u(x, y, t) |

, m−1 < α < m.

t vα

i=0

∂tk

t=0

(1)

Based on equation (1) above, the correct version of equation (5) is:

m−1 k

 1 E(u(x, y, t)) − Σ u2+i−α ∂ u(x, y, t) | = E(u (x, y, t) + u (x, y, t))

E(u(x, y, t)) = v2u(x, y, 0) + vα (E(uxx(x, y, t) + uyy(x, y, t))) . (2)

Therefore, the authors must modify section 3 and section 4 carefully as suggested in the first review.

Updated as required as suggested with right result of Elzaki transform.

---

## [Decision Letter · Decision Letter 2]

18 Jan 2024

Elzaki residual power series method to solve fractional diffusion equation

PONE-D-23-29441R2

Dear Dr. Emadifar,

We’re pleased to inform you that your manuscript has been judged scientifically suitable for publication and will be formally accepted for publication once it meets all outstanding technical requirements.

Kind regards,

Luan Carlos de Sena Monteiro Ozelim, D.Sc.

Academic Editor

PLOS ONE

Additional Editor Comments (optional):

Reviewers' comments:

Reviewer's Responses to Questions

**Comments to the Author**

1. If the authors have adequately addressed your comments raised in a previous round of review and you feel that this manuscript is now acceptable for publication, you may indicate that here to bypass the “Comments to the Author” section, enter your conflict of interest statement in the “Confidential to Editor” section, and submit your "Accept" recommendation.

Reviewer #2: All comments have been addressed

2. Is the manuscript technically sound, and do the data support the conclusions?

Reviewer #2: Yes

3. Has the statistical analysis been performed appropriately and rigorously? 

Reviewer #2: Yes

4. Have the authors made all data underlying the findings in their manuscript fully available?

Reviewer #2: Yes

5. Is the manuscript presented in an intelligible fashion and written in standard English?

Reviewer #2: Yes

6. Review Comments to the Author

Reviewer #2: No further comments. The authors carefully modified the manuscript. The revised manuscript can be accepted for publication.

7. PLOS authors have the option to publish the peer review history of their article (what does this mean?). If published, this will include your full peer review and any attached files.

Reviewer #2: **Yes: **Shehu Maitama

---

## [Editor Report · Acceptance letter]

2 Feb 2024

PONE-D-23-29441R2 

PLOS ONE

Dear Dr. Emadifar, 

I'm pleased to inform you that your manuscript has been deemed suitable for publication in PLOS ONE. Congratulations! Your manuscript is now being handed over to our production team.

Kind regards, 

on behalf of

Dr. Luan Carlos de Sena Monteiro Ozelim 

Academic Editor

PLOS ONE